# Best Practices in the Management of *Clostridioides difficile* Infection in Developing Nations

**DOI:** 10.3390/tropicalmed9080185

**Published:** 2024-08-19

**Authors:** Rafael Mendo-Lopez, Carolyn D. Alonso, Javier A. Villafuerte-Gálvez

**Affiliations:** 1Division of Infectious Disease, University Hospitals Cleveland Medical Center, Cleveland, OH 44106, USA; 2School of Medicine, Case Western Reserve University, Cleveland, OH 44106, USA; 3Louis Stokes Cleveland VA Medical Center, Cleveland, OH 44106, USA; 4Division of Infectious Disease, Beth Israel Deaconess Medical Center, Boston, MA 02215, USA; calonso@bidmc.harvard.edu; 5Harvard Medical School, Harvard University, Boston, MA 02215, USA; jvillaf1@bidmc.harvard.edu; 6Division of Gastroenterology, Beth Israel Deaconess Medical Center, Boston, MA 02215, USA

**Keywords:** *Clostridioides difficile*, developing countries, Africa, Asia, Latin America, Europe

## Abstract

*Clostridioides difficile* infection (CDI) is a well-known cause of hospital-acquired infectious diarrhea in developed countries, though it has not been a top priority in the healthcare policies of developing countries. In the last decade, several studies have reported a wide range of CDI rates between 1.3% and 96% in developing nations, raising the concern that this could represent a healthcare threat for these nations. This review defines developing countries as those with a human development index (HDI) below 0.8. We aim to report the available literature on CDI epidemiology, diagnostics, management, and prevention in developing countries. We identify limitations for CDI diagnosis and management, such as limited access to CDI tests and unavailable oral vancomycin formulation, and identify opportunities to enhance CDI care, such as increased molecular test capabilities and creative solutions for CDI. We also discuss infection prevention strategies, including antimicrobial stewardship programs and opportunities emerging from the COVID-19 pandemic, which could impact CDI care.

## 1. Introduction

*Clostridioides difficile* is a spore-forming gram-positive bacterium producing two homologous toxins, essential in the disease etiopathogenesis: toxin A (TcdA) and toxin B (TcdB) [1]. *C. difficile* toxins are necessary, yet not sufficient, to trigger the activation of the pyroptotic cascade through a signaling axis including the innate pathogen recognition receptor Toll-like receptor 9 (TLR9), and dependent on the NLR3-ASC inflammasome cascade. TcdA and TcdB have a glucosyl-transferase activity on host intracellular Rho-GTPases leading to their inactivation; this additionally can trigger not only intracellular pathogen sensing mechanisms aiding inflammasome activation, but also cytoskeletal disruption resulting in a dissociation of the tight junctions between colonocytes, loss of epithelial integrity, and cell death [2,3].

Currently, *C. difficile* is responsible for most cases of hospital-acquired infectious diarrhea in the United States (U.S.). Furthermore, CDI is no longer thought to be restricted to healthcare facilities and is known to occur in community settings, resulting in a significant healthcare challenge in the U.S. [4]. A 2020 U.S. Centers for Disease Control and Prevention (CDC) report, which included over 10 CDC Emerging Infection Programs (EIPs), established a CDI incidence rate of 101.3 cases per 100,000 individuals [5]. Meanwhile, a 2024 European Centre for Disease Prevention and Control annual report, including 26 countries/administrations and covering 2016–2020, established a crude CDI incidence density between 1.94 and 3.16 cases per 10,000 patient-days [6]. Beyond its frequency, CDI causes significant burdens on healthcare systems due to its associated morbidity, mortality, and cost. CDI has a reported mortality rate of 6–10% and 3.5%, in the U.S. and Europe, respectively [6,7]. In addition, two studies have estimated the healthcare costs of CDI to range from $6188.67 to $24,205 [8,9]. These studies primarily focused on data from the U.S. and Europe and may not reflect the global experience. Over the last twenty years, multiple studies have demonstrated that CDI is a global healthcare challenge [10]. It is essential to recognize that the epidemiology, diagnostic methods, and conventional therapy choices may differ between developed and developing nations due to various factors, such as demographic variations, testing and treatment resources and algorithms, and unique features of each healthcare system.

This review aims to provide an updated overview of the CDI epidemiology, diagnostics, and treatment literature available from developing nations and comment on them based on the authors’ experiences. HDI is a concise metric developed by the United Nations Development Program, comprising three dimensions of human development: long and healthy life, knowledge, and a decent standard of living. Countries are classified based on HDI as having very high (HDI > 0.8), high (HDI 0.7–0.79), medium (HDI 0.55–0.69, or low (HDI < 0.55) human development. We arbitrarily defined developing countries as having a Human Development Index (HDI) below 0.8 in order to include as many developing countries as possible [11,12].

## 2. CDI in Developing Nations: Under-Recognized, Under-Measured

### 2.1. Current Epidemiology

CDI epidemiology data from developing countries consist of cross-sectional studies conducted in various healthcare settings reporting point prevalence or positivity rates (see Box 1). However, no population-based studies have established the incidence or prevalence of CDI in these nations. In Figure 1 and Table A1, we report the median CDI rate for each country and CDI rates, respectively. Data on *C. difficile* ribotypes are listed in Table A2, and key abbreviations are summarized in Abbreviations/Acronyms section below.

Box 1Definitions.-**Point prevalence =** Number of current cases at specific point in time/Population at the same specified point of time-**Positivity rate =** Number of subjects who test positive for a test/Total subjects tested

#### 2.1.1. Africa

A comprehensive review reported a CDI frequency range from 1.03% to 92.38% [13]. It predominantly included studies from Egypt, Kenya, and South Africa. However, other countries, such as Cote d’Ivoire, Ghana, Nigeria, Central African Republic, Malawi, Tanzania, Zambia, Zimbabwe, and Botswana, were also included. A study from Kenya reported the highest CDI frequency at 92.38% using combined toxinogenic culture and polymerase chain reaction (PCR)-based methods for CDI diagnosis [14]. A study from the Central African Republic had the lowest CDI rate of 1.03%, utilizing cell-culture cytotoxicity neutralization assay (CCNA) for CDI diagnosis [13]. Despite substantial efforts to conduct CDI studies and elucidate its distribution in Africa over the past decade, there is still a paucity of published reports. The absence of reports from populous countries, such as Ethiopia (West Africa), the Democratic Republic of Congo (Central Africa), and Sudan and Morocco (North Africa), limits the wide generalizability of these reports to the African continent.

#### 2.1.2. Asia

Most published studies from Asia come from China. However, other developing countries have also conducted CDI studies. A systematic review and meta-analysis included Asian studies between 2000 and 2016, focusing on CDI prevalence. Its findings revealed a pooled *C. difficile* positivity rate of 14.8%, with healthcare-associated CDI (HA-CDI) of 16.4%, and community-associated CDI (CA-CDI) of 5.3%. Additionally, studies including inpatients and outpatients had a pooled *C. difficile* positivity rate of 11.1% [15]. This review included 51 studies: 31.4% from China, 33.3% from other developing nations (Iran, Lebanon, India, and Pakistan), and 35.3% from developed nations (Japan, Korea, Qatar, Singapore, Thailand, and Malaysia). Of all the included studies performed in developing countries, the CDI frequency rate distribution ranged from 4.2% to 61.4%. The lowest CDI rate was observed in a hospital-based Chinese study that utilized combined toxin B PCR and culture for CDI diagnosis (4.2%). The highest CDI rate was reported in a hospital-based Lebanese study that utilized a toxin A/B enzyme immunoassay (EIA) or CCNA culture for CDI diagnosis (61.4%). The remaining studies from other Asian developing countries found wide variation in CDI rates (see Table A1).

#### 2.1.3. Latin America

A 2022 comprehensive review revealed a rising CDI rate in the region, with rates in developing countries ranging from 4.5% to 96% (see Table A1). The lowest CDI rate of 4.5% corresponds to a Brazilian study that utilized a combined a glutamate dehydrogenase (GDH)/toxin A/B rapid EIA and toxin A/B PCR. Meanwhile, the highest CDI rate (96%) was found in a Mexican study that used either a toxin A/B EIA or toxin A/B PCR to establish the CDI diagnosis. Moreover, this review included some developed countries, such as Argentina, Chile, and Costa Rica, reporting rates between 6.5% and 86% [16]. 

Most studies have been cross-sectional and have been performed predominantly in Brazil and Mexico. However, other countries, such as Peru, Colombia, and Paraguay, have also published studies related to CDI epidemiology (see Table A1) [17,18,19,20,21,22]. Researchers from Cuba, Honduras, and Ecuador have published case reports [16].

#### 2.1.4. Europe

Most published studies from Europe correspond to developed countries such as Germany, Spain, Italy, Portugal, Ireland, etc. However, Albania, Armenia, Azerbaijan, Bosnia and Herzegovina, Bulgaria, Moldova, North Macedonia, and Ukraine have HDIs below 0.8. Among those, only four countries had reported CDI data. In Ukraine, two case reports were published. One case of recurrent CDI was in a 5-year-old child who had a successful treatment outcome, and the other study was a CDI case in a 65-year-old male who ended up with a mortality outcome. 

A North Macedonian study found a CDI rate of 13.2% (182 of 1380 fecal samples) [23]. Another two studies reported the most frequent CDI ribotypes (see Table A2) [24,25]. Similarly, a Bulgarian study reported a lower HA-CDI rate of 3.3% by using a rapid immunochromatographic test for CDI diagnosis [26]. Another Bulgarian retrospective cohort study compared two groups (pre-COVID vs. COVID) and found an increase in the CDI rate of 21.95% in the COVID group [27].

Finally, a retrospective study from Bosnia and Herzegovina reported a CDI frequency of 35.1% in a 4-year study by establishing CDI diagnosis with toxin A/B enzyme-linked immunosorbent assay (ELISA) [28]. 

### 2.2. Limited Awareness

Limited awareness of CDI as a potential cause of diarrhea may be a contributory factor in under diagnosis of CDI in developing nations. A 2018 South African study aimed to identify the barriers and facilitators to providing quality care for CDI. It explored the perceptions and practices among physicians, nurses, and pharmacists. Their knowledge assessment consisted of a semi-structured interview divided into three sections: CDI identification, diagnosis, and treatment/prevention [29]. The study revealed a lack of or limited knowledge about CDI, particularly among nurses and pharmacists. The study also reported that CDI receives less urgency in South Africa due to competing attention from other more prevalent diseases, such as tuberculosis and HIV. Regarding CDI diagnosis, the study identified several barriers, including difficulty or delay in stool sample collection due to staff shortages or non-standardized laboratory sample collections. Other barriers included delayed test results, variable perceptions on time to result, test costs, and lack of physician automated notification systems. For CDI management/prevention, the main barriers identified were the inaccurate route of administration of treatment, delayed treatment due to time gaps between test ordering and result review, limited availability or prioritization of isolation rooms for multidrug-resistant tuberculosis (MDR-TB), inconsistent hand hygiene, and inconsistent use of supplies (gowns and gloves) despite availability. 

Similar barriers have been observed by authors RML and JVG during medical training in Peru, where they have noted limited CDI testing availability and inappropriate CDI testing. This last barrier was studied in a U.S. academic hospital, and it was recognized that the lack of documentation of diarrhea, reported either by the patient or nurse, in the medical record was the main reason for inappropriate testing, followed by the perception of the primary team that the patient was at high-risk of developing CDI [30].

Furthermore, comparable barriers have been found in developed countries. A 2024 Saudi Arabian questionnaire-based cross-sectional study reported a knowledge gap among healthcare workers for CDI diagnosis, management, and severity classification. It found that 35% of healthcare workers were unsure about their CDI diagnosis algorithm at their institution, and only 27.9% of recruited healthcare workers demonstrated an adequate CDI knowledge (>70% of corrected answers) [31].

### 2.3. A High-Risk Environment: Aging Population, Inadequate Antibiotic Stewardship

The main risk factors for CDI include advanced age (≥65 years old), hospitalization (current or prior hospitalization, prolonged length of stay), antibiotic or chemotherapy exposure, and chronic digestive or systemic diseases (i.e., inflammatory bowel disease, renal failure requiring renal replacement therapy). Other risk factors include malignancy and immunosuppression [32,33,34,35].

Developing countries have experienced or are currently undergoing an epidemiologic transition characterized by changes in population growth trajectories and composition. These changes involve shifts in age distribution, specifically from younger to older populations, and modifications in mortality patterns, including increased life expectancy and a reordering of the relative importance of different causes of death [36]. According to the United Nations database, the population aged 60 years and above in Africa, Asia, Latin America, and the Caribbean nearly doubled between 2001 and 2021 [37]. The aging population in these regions may contribute to a higher incidence of CDI. The elderly population is at higher risk of developing CDI not solely due to their age but also due to age-related factors, such as an immune senescence, higher rates of organ dysfunction, and increased malignancy rates [38,39].

In some developing countries, inadequate antibiotic stewardship and infection control measures may contribute to the rise and spread of CDI. Overutilization of antibiotics by healthcare professionals and widespread access to antimicrobials in retail pharmacies without a prescription are among the factors that may contribute to a CDI increase [40,41]. A major challenge in implementing infection control measures is a shortage of private rooms to isolate patients due to a lack of appropriate hospital infrastructure. For example, authors RML and JVG report that public hospitals in Peru typically have large 20–40 bed open wards or 4–6 patient shared rooms, with limited isolation rooms often reserved for airborne precautions (MDR-TB, SARS-CoV-2). Insufficient access to gowns and gloves is another challenge affecting the implementation of contact precautions. In addition, high patient-to-clinician ratios and scarcity of hand hygiene stations may hinder appropriate hand hygiene practices, thereby disseminating CDI [41].

The paucity of established antibiotic stewardship programs (ASPs) in hospitals and outpatient settings is also a concern. While efforts are being made to develop ASPs in some countries (India, Brazil, Colombia, and South Africa), concerns remain about the widespread availability of over-the-counter antimicrobials in certain countries (China, Brazil, India, Mozambique, Ghana, South Africa, Vietnam, Bangladesh, Mexico, Peru, etc.) [41,42,43]. Economic limitations also hinder the development of appropriate staffing and resources for ASPs. Additionally, there is a knowledge gap about antimicrobial prescribing among clinicians, with studies reporting high rates of unnecessary antibiotic usage and identifying potential determinants for overutilization of antibiotics (demand for antibiotics from patients, lack of supporting tests, and limited capacity of primary healthcare prescribers) [44,45]. The combination of these factors, along with an aging population and inadequate infection control measures, contributes to a higher risk population and more frequent opportunities to acquire CDI in developing countries.

## 3. Limited Diagnostic Capabilities

### 3.1. State of the Art in CDI Diagnostics

The 2017 IDSA/SHEA guidelines recommend a multistep algorithm for CDI diagnosis based on either a nucleic acid amplification test (NAAT) alone (for pre-established institutional criteria for stool submission) or a toxin A/B EIA as part of multistep diagnosis in combination with GDH and/or a NAAT (i.e., a combined GDH and toxin A/B EIA, GDH and toxin A/B EIA arbitrated by NAAT, or combined NAAT and toxin A/B EIA). Also, they recommended that CDI testing be performed in a patient with new-onset and unexplained diarrhea defined as three or more unformed stools in 24 h [46].

The 2016 European Society of Clinical Microbiology and Infectious Diseases (ESCMID) CDI diagnosis guidance document proposes a 3-step algorithm as an alternative option for CDI diagnosis, starting with either a NAAT or GDH test followed by a toxin A/B EIA for positive results. If the toxin A/B EIA is negative, the third step involves performing a NAAT (in case the first test was GDH) or toxinogenic culture to confirm a CDI [47].

South African CDI guidelines have mirrored the IDSA/SHEA diagnostic recommendations [48]. Meanwhile, the Mexican Consensus for CDI diagnosis and treatment recommends CDI diagnosis be performed in a two-step approach, starting with GDH or a NAAT and followed by a toxin A/B EIA if the first test is positive; or a multiple-step approach, starting with GDH and a toxin A/B EIA and followed by a NAAT if the GDH is positive and a toxin A/B EIA if negative [49].

India does not have integrated criteria for CDI diagnosis. A systematic literature review on CDI burden revealed that CDI diagnosis was established with single or multiple tests. The main tests were anaerobic or toxinogenic cultures and ELISA [50]. Numerous studies from various countries (Bangladesh, Algeria, Iran, Lao, Lebanon, Iran, and India) have employed ≥2 CDI tests: toxin A/B EIA, GDH, toxinogenic culture, and toxin A/B PCR [51,52,53,54,55,56]. Meanwhile, other countries have established CDI diagnosis based on a single test: Nigeria, Kenya, and Iran use toxin A/B EIA; whereas Pakistan uses CCFA culture [57,58,59,60]. An Egyptian study reported CDI not routinely being detected in most hospitals, raising concern about how CDI diagnosis is established in these facilities [61].

In summary, the diagnostic approach to CDI remains heterogeneous in developing countries, limiting the understanding and recognition of CDI within each country. This is a reflection of a major gap in knowledge and technology in CDI diagnosis, which is also present in developed countries [62]. However, it also represents an opportunity for local and pragmatic innovation in diagnostics by researchers in developing nations.

### 3.2. A Paucity of Clinical Microbiology Infrastructure

One of the main challenges in developing countries is the limited access to appropriate laboratory resources for detecting *C. difficile*. Diagnostic tests, such as GDH, toxin A/B EIA, and NAAT, are often unavailable or too expensive, resulting in underdiagnosed CDI cases and an unknown burden of disease. To establish CDI diagnostic algorithms that are realistic and accurate for their own needs, developing countries must overcome this challenge. An analysis of pathology and laboratory medicine (PALM) services found different barriers being the key components: inadequate infrastructure, lack of training and education, insufficient human resources, and lack of established quality standards and accreditation [63].

The challenge extends beyond physical laboratory infrastructure and encompasses technical support for instrumentation, supply chains, information technology (IT), and integrated systems [63,64]. For example, the Peruvian Ministry of Health system is still mainly based on paper records, resulting in inefficiency and errors. Moreover, most Peruvian Ministry of Health tertiary hospitals do not have in-house CDI tests and must send samples to private external laboratories requiring out-of-pocket pre-payment, erecting barriers to testing and resulting in delays in diagnosis and treatment.

In addition to infrastructure, ensuring accuracy and reproducibility of test results is crucial. Developed countries adhere to strict regulations overseen by national governments or independent accreditation bodies, with professional societies playing a role in their development, implementation, and updates. In contrast, developing countries may lack regulations and if they exist they may vary widely between institutions, regions, and countries. Some countries, such as Kenya, and Malaysia, have developed in-country standards and accreditation systems to overcome this problem [63]. The World Health Organization (WHO) has also developed a handbook providing comprehensive guidance on laboratory quality management systems [65]. However, these efforts have not been sufficient, as evidenced by a study showing that 75.5% (37/49) of laboratories in Sub-Saharan African countries fail to meet international quality standards [66].

Ultimately, the insufficient human resources/workforce capacity and lack of training and education are interconnected components. The former pertains to a laboratory’s human resources, while the latter relates to the knowledge and training of the professionals. In Africa, a study demonstrated how the shortage and maldistribution of healthcare workers, including laboratory technicians, remained a challenge to attainment of universal access to health services [67].

### 3.3. Unexpected Opportunities

The COVID-19 pandemic has brought attention to the importance of PALM, particularly in clinical microbiology, as an essential component of healthcare. It also showed a general lack of preparedness for major public health crises, in developed and developing countries alike. It is to be noted that some African countries defied grim expectations on their initial response to the pandemic [68]. The pandemic has also spurred development in the microbiological field in many developing countries due to an unprecedented and urgent need to strengthen PALM capabilities [69,70,71].

For instance, Peru expanded its molecular testing capacity exponentially by acquiring thermocyclers during the COVID-19 pandemic [72]. Similarly, in India, the number of COVID-19 testing laboratories increased from 14 to 1596 in the span of 6 months between February and August 2020 [69]. This rapid expansion was in part supported by governments but mostly by private sector initiatives, which should not be overlooked as potential partners to expand testing for CDI. These opportunities translated into expanding molecular testing infrastructure in developing countries, which could be leveraged for PCR testing for CDI, especially now that the peak demand for SARS-CoV-2 testing has declined and some unused testing capacity exists. This could contribute to a better understanding of CDI epidemiology in developing countries.

Initiatives for PALM development were already underway prior to the pandemic. Multiple proposals, such as the creation of Strengthening Laboratory Management Toward Accreditation (SLMTA), Stepwise Laboratory Improvement Process Towards Accreditation (SLIPTA), Africa Centers for Disease Control and Prevention (Africa CDC), and various other worldwide efforts, have aimed to promote laboratory capacity improvement in developing countries [64]. Notable examples include the implementation of molecular testing for COVID-19 in Timor-Leste and the accreditation of laboratories to international standards in 49 countries outside of South Africa [64,73]. Public–private partnerships have also been established, such as those in Kenya, Ethiopia, Mozambique, and Uganda, where international institutions collaborated to strengthen laboratory systems and reduce turnaround times [74].

Finally, point-of-care (POC) testing, which is also available for CDI using GDH, toxin EIA, and NAAT, has gained prominence in the last decade [75]. POC testing can help address challenges related to infrastructure limitations by reducing the burden in the supply chain [74,75]. It can also improve turnaround time and overcome issues with insufficient human resources by utilizing task shifting/sharing within the existing healthcare workforce. A study exemplified implementing a CDI POC toxin B PCR (Cepheid GeneXpert system) in a London hospital (in three medical wards and two intensive care units) to assess the acceptability, ease to use, change in turnaround time, and clinical utility. This study concluded that CDI POC toxin B PCR testing using the GeneXpert system was feasible and acceptable among the nursing staff and laboratory technicians, who performed the test in the medical ward and intensive care unit, respectively, and also decreased the turnaround time from 18 h to 1.85 h [75].

These new advancements in microbiology laboratories in developing countries create an opportunity to redirect resources toward accurately determining the CDI epidemiology and establishing CDI diagnostic algorithms applicable to daily patient care.

## 4. Challenges of Treatment

### 4.1. Metronidazole, Always Available but No Longer SOC

The 2017 IDSA/SHEA guidelines no longer recommend metronidazole as the standard of care (SOC) for CDI management and this was reaffirmed in the updated 2021 IDSA/SHEA and 2021 ESCMID guidelines focused on CDI management. However, it is still considered an alternative option for non-severe CDI if fidaxomicin or oral vancomycin are unavailable [46,76,77]. The 2018 Mexican Consensus for CDI Prevention, Diagnosis, and Treatment, and the 2020 South African Society of Clinical Microbiology CDI guideline adopt metronidazole as an alternative treatment if oral vancomycin is not available (see Table 1) [48,49]. The 2020 Taiwan CDI guidelines recommend metronidazole 500 mg three times daily for a 10-day course for the first non-severe CDI episode as one treatment option [35]. This shows variability in guidance which could reflect different speeds of practice change implementation, pragmatic reasons such as local drug availability or pricing, or even local differences in outcomes with different drugs.

CDI guidelines from other developing countries are lacking, and in many if not most regions metronidazole is still considered the first line treatment in non-severe CDI. One such example comes from a CDI survey performed in the Asia-Pacific region, including 40 sites, including developed countries (Japan, Australia, Malaysia, Singapore, Thailand, and Republic of Korea) and developing countries (China, India, Indonesia, Philippines, and Vietnam). In this study, 54.2% of recruited patients were from developing countries. Among all the non-severe CDI patients, 92.6% received metronidazole treatment [78]. Another study from India found 43% of CDI patients were treated with single agent metronidazole; 34% were treated with combined metronidazole and vancomycin [79]. Similar findings were observed in studies from Iran (57.9% and 28.9%, respectively) and Mexico (35.3% and 48.2%, respectively) [80,81].

In summary, metronidazole still has a de facto prominent role in the SOC for primary CDI management in developing countries. Reasons include the availability of oral vancomycin, the lack of access to other current SOC treatments, and/or current SOC treatment costs. Further studies are needed to evaluate the efficacy of treatment regimens in developing regions in comparison to the available published literature.

### 4.2. Oral Vancomycin: Creative Solutions

The updated 2021 IDSA/SHEA CDI guideline suggests fidaxomicin (or oral vancomycin as an alternative) as the first line treatment for CDI. In these guidelines, vancomycin may be used for primary CDI, the first or subsequent CDI recurrence, or fulminant CDI (see Table 1) [46]. One limitation to use of oral vancomycin in developing countries is a limited access to oral vancomycin capsule formulations [82].

Some creative solutions have been made, including the use of a formulation of vancomycin intended for intravenous (IV) administration as vancomycin administered orally, as reported in the 2018 Mexican Consensus for CDI Prevention, Diagnosis, and Treatment [49]. That guideline recommends mixing a 500 mg vial of IV vancomycin with 10 mL of sterile water and giving only 2.5 mL, equivalent to 125 mg, every 6 h. The guideline also recommends administering the vancomycin with juice to improve tolerance [49].

Another study evaluated the stability of vancomycin solution prepared from reconstituting commercially available vancomycin intravenous vials with sterile water for injection and mixed with an Ora-Sweet vehicle and distilled water [83]. This study demonstrated that this vancomycin solution of 25 mg/mL could remain stable for 75 days if stored at 4 °C. However, it is important to highlight that this oral vancomycin solution has a more complex preparation than the one described in the Mexican study.

Finally, a study has compared the clinical cure rate (CCR) of CDI between an oral vancomycin capsule and an oral vancomycin solution (prepared from vancomycin powder for injection) given for initial severe CDI treatment [84]. It showed no significant difference in the CCR between the oral vancomycin capsule and oral vancomycin solution at day 10 (59% and 64%, respectively). This change to solution formulation could reduce the 10-day oral vancomycin treatment cost from $1400 to $100 for a primary CDI treatment (125 mg 4 times daily).

This resourceful solution to implement oral vancomycin in developing countries may be critical for decreasing treatment failure and rates of recurrence. 

### 4.3. Challenges to Fecal Microbiota Transplantation

Fecal microbiota transplantation (FMT) is considered an effective treatment option for patients with recurrent, refractory, or fulminant CDI (see Table 1) [46,77].

Mexican, South African, and Taiwan guidelines have also incorporated FMT as part of recurrent CDI (rCDI) management [35,48,49]. Several studies from Brazil, Mexico, South Africa, China, Colombia, and India reported successful FMT for refractory and recurrent CDI [85,86,87,88,89].

The high efficacy and relative low cost of FMT makes it attractive as a therapeutic option in the developing world. However, several challenges need to be addressed to expand the use of FMT in developing countries. These include the paucity of regulatory frameworks, guidelines, standardized screening of donors for infections, storage, and administration of FMT [90].

The main barrier currently is the local availability of safe donor material. To the best of our knowledge, the largest repository of donor fecal matter is Openbiome, located in the United States [91]. However, various other stool banks have emerged across the globe, such as the Asia Microbiota Bank in China, Leiden University Medical Center in the Netherlands, PHE Public Health Laboratory and Portsmouth Hospitals in the United Kingdom, Saint-Antoine Hospital in France, University Hospital College in Germany, Hospital Ramon & Cajal in Spain, and Medical University Graz in Austria [92]. Notably, all these facilities are situated in developed nations, except for the Asia Microbiota Bank. This disparity underscores a significant limitation for FMT implementation in developing countries [93]. The creation of new stool banks in developing countries or the establishment of international partnerships with existing repositories to provide fecal donor material for FMT will play a key role if FMT is to be widely used in the developing world.

### 4.4. Challenges to Introducing Newer Therapeutics: Monoclonal Antibodies, Newer Antibiotics, Bacteriophage

While fidaxomicin and bezlotoxumab (an anti-toxin B monoclonal antibody) have joined the standard of care to prevent rCDI in the latest IDSA/SHEA and ESCMID guidelines (see Table 1), challenges to their implementation remain in both developed and developing countries [76,77].

The current costs of fidaxomicin and bezlotoxumab are approximately $883.60–$4335 and $3896, respectively [94,95]. This has been the most substantial barrier to implementation in the U.S. and European healthcare systems [94,95]. However, despite the high initial cost, numerous studies suggest both treatments may be cost-effective based on reductions in recurrence rates and subsequent hospitalizations [94,95,96,97,98,99]. Pricing adjusted to local cost-effectiveness thresholds or, within a few years, generic or biosimilar versions may expand availability in developing nations. 

Also, developing countries with universal healthcare for their populations, such as Brazil or China, may find objective benefits in implementing these new therapies [100,101]. However, other aspects such as drug registration and negotiation between the government or healthcare institutions and drug manufacturers should be considered to establish reasonable pricing and appropriate distribution.

In the past year, two live biotherapeutic products (LBPs) designed to restore gut microbial balance and function have been approved by the U.S. Food and Drug Administration. RBX-2660 (Rebyota), a rectally-administered microbiota suspension prepared from human stools demonstrated safety and modest effectiveness for prevention of recurrent CDI in adults following standard-of-care treatment [102]. SER-109 (Vowst), another donor stool-derived LBP consisting of oral capsules containing purified Firmicutes spores, has demonstrated superior effectiveness in reducing the risk of CDI recurrence compared to placebo following standard antibiotic treatment (oral vancomycin or fidaxomicin) [103]. The approval of both LBPs will likely disrupt the market putting in question the need for FMT; though refractory and fulminant CDI remain unmet needs. Uptake and real-world implementation of these LBPs remains to be determined, even in North America.

Narrow spectrum CDI antibiotics remain another fertile field of investigation with ridinilazole (3 phase-III studies) and CRS3123 (phase-II study recruiting) being promising clinical phase assets [104,105,106,107,108]. However, if they were to be approved, their implementation in developing countries may face similar challenges to fidaxomicin. 

### 4.5. Opportunity: Enhanced Antimicrobial Stewardship

Antimicrobial stewardship involves the development of systematic measures to optimize antimicrobial use, decrease unnecessary antimicrobial exposure, and decrease the emergence and spread of antibiotic resistance [109]. These measures directly influence CDI epidemiology by improving infection prevention and control measures and promoting appropriate treatment strategies. A systematic review showed a decrease in CDI incidence with the implementation of ASP; an incidence ratio of 0.68 (95% CI 0.53–0.88) was observed with an estimated mean protective effect of 32% [110].

Despite limited resources and infrastructure, some developing countries in Latin America, Asia, and Africa have demonstrated that it is possible to implement effective antibiotic stewardship programs (ASP) to combat antimicrobial resistance and improve the quality of care for infectious diseases, such as CDI [111]. South Africa has implemented a national ASP in public hospitals since 2015. Brazil also made significant strides in ASP implementation, with the Brazilian Antimicrobial Stewardship Program (PROA) established in 2016. Meanwhile, Colombia developed a Nosocomial Resistance Study Group to surveil resistance patterns in 32 public and private hospitals through 11 cities. Similarly, India implemented the National Action Plan on Antimicrobial Resistance in 2017. China, Indonesia, Egypt, and the Philippines, have also successfully implemented ASPs [109,111].

The WHO has elaborated a framework for developing national action plans as a first step for creating local ASPs [112]. International efforts have taken place, such as the U.S. Center for Disease Dynamics and Economics and Policy (CDDEP)-supported Global Antimicrobial Resistance Partnership (GARP), a program providing tools for developing countries to create their own antibiotic use and resistance databases and supporting stewardship activities. Other resources available include ReAct, an open-access web-based platform to create a national action plan supported by the Swedish International Development Cooperative Agency and Uppsala University [109,111].

It is imperative to continue implementing ASPs in developing countries; they not only optimize the use of limited resources and reduce antibiotic resistance but have a major impact on other major problems such as CDI.

## 5. Key Facts

Despite challenges in establishing CDI epidemiology, diagnosis, and management/prevention in developing countries, clinicians and researchers from these areas have been making substantial efforts to bridge this gap in the past 20 years. We report a wide variability in CDI rates in developing nations owing to the design of most studies favoring cross-sectional single-center studies over population-based longitudinal ones, as well as heterogeneous testing methods. Numerous barriers to optimal CDI care are described including knowledge gaps, competing public health priorities, paucity of infrastructure, and high costs of testing and ideal treatment. However, opportunities for improvement abound, including oral use of intravenous vancomycin formulations, expanded molecular diagnostic infrastructure brought about by the COVID-19 pandemic, increasingly available POC tests, and the development of ASPs. The establishment of best practices in the diagnosis, management, and prevention of CDI in developing nations will need coordinated efforts from patients, government, clinicians, and the private sector to generate standards and guidance and allocate funds wisely. 

## Figures and Tables

**Figure 1 tropicalmed-09-00185-f001:**
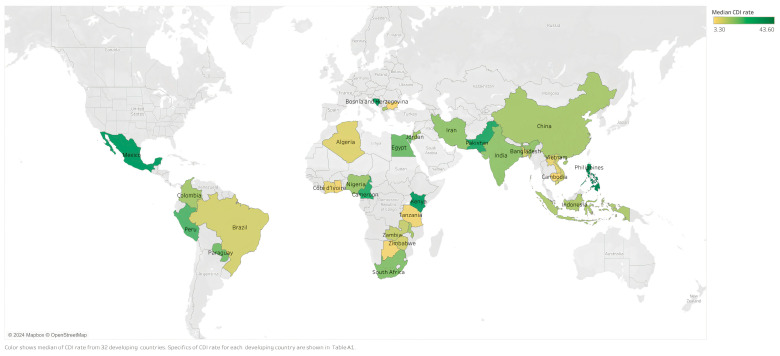
Epidemiological distribution of *Clostridioides difficile* infection in developing countries.

**Table 1 tropicalmed-09-00185-t001:** Comparison of *Clostridioides difficile* infection management guidelines.

CDI	IDSA/SHEA	ESCMID ^a^	South African Society of Clinical Microbiology ^b^	Taiwan Guidelines	Mexican Consensus
**Primary episode, non-severe**	**Preferred regimen:** SOC ^c^ FDX				
	**Alternative regimen:** SOC ^c^ VAN		**Preferred regimen:** SOC ^c^ VAN		
	**Alternative regimen:** SOC ^c^ MET ^d^			**Preferred regimen:** SOC MET	**Alternative regimen:** SOC MET ^d^
**Primary episode, severe ^e^**	**Preferred regimen:** SOC FDX				
	**Alternative regimen:** SOC VAN	**Preferred regimen:** SOC VAN			**Preferred regimen:** VAN 125–250 mg qid for 14 days
		**Oral administration not possible ^f^:** Rectal or nasoduodenal delivery +/− adjunctive IV MET 500 mg tid		**Alternative regimen:** TEC 200 mg bid for 10 days	
**First recurrence**	SOC FDX OR FDX EPX ^g^	SOC FDX (if primary episode was treated with VAN or MET)	SOC FDX (if VAN was used for primary episode)	SOC FDX (if VAN was used for primary episode in a patient without risk factors ^h^)	
	**Alternative regimen:** VAN prolonged ta-pered and pulsed regimen ^i^ OR SOC VAN	SOC VAN OR SOC FDX + BEZ 10 mg/kg IV once (if primary episode treated with FDX)	SOC VAN (if MET was used for primary episode)	SOC VAN (if MET was used for primary episode in a patient without risk factors ^h^)	SOC VAN (if MET was used for primary episode)
	**Adjunctive therapy:** BEZ 10 mg/kg IV once after SOC	VAN prolonged tapered and pulsed regimen (if FDX or BEZ are unavailable)	VAN prolonged tapered and pulsed regimen (if standard VAN was used in primary episode)	TEC 100–200 mg bid for 10 days (if VAN was used for primary episode in a patient without risk factors). **Patient with risk factors ^h,j^:** VAN extended-regimen ^k^ OR FDX EPX ^g^	VAN prolonged tapered and pulsed regimen (if SOC VAN was used in primary episode)
**Second or subsequent recurrence**	SOC FDX	SOC FDX + BEZ 10 mg/kg IV once OR SOC FDX followed by FMT	SOC FDX	FDX EPX if it was not previously used	
	VAN tapered and pulsed regimen OR SOC VAN followed by RAX 400 mg tid for 20 days	SOC VAN followed by FMT OR SOC VAN + BEZ 10 mg/kg IV once	VAN prolonged tapered and pulsed regimen	VAN extended regimen ^k^	VAN tapered and pulsed regimen OR VAN 125 mg qid for 10–14 days followed by RAX (unavailable in Mexico) 400 mg tid for 20 days
	**Adjunctive therapy:** BEZ 10 mg/kg IV once after SOC	**Alternative regimen:** VAN tapered and pulsed regimen (if FDX, BEZ, and FMT are unavailable)		TEC 100–200 mg bid for 10–14 days if it was not previously used	
	FMT	FMT	FMT (for third and subsequent recurrence)	FMT	FMT
**Fulminant CDI ^l^**	VAN 500 mg qid PO or NGT + IV MET 500 mg tid	SOC VAN OR SOC FDX + surgical consultation	VAN 500 mg qid PO or NGT + IV MET 500 mg tid	VAN 125–500 mg qid PO or NGT + IV MET 500 mg tid	VAN 250–500 mg qid PO or NGT + IV MET 500 mg tid
	If ileus present: Consider adding rectal VAN 500 mg in 100 mL NaCl qid as retention enema.		If ileus present: Consider adding rectal VAN 500 mg in 100 mL NaCl qid as retention enema.	VAN 125–500 mg qid PO or NGT plus VAN 0.25–1 g bid-qid per rectum	If ileus or abdominal distention: VAN 500 mg qid rectal

**Notes:** ^a^ Treatment for high-risk recurrence in primary non-severe CDI: FDX 200 mg bid for 10 days or FDX 200 mg bid for 5 days followed by once qod for 20 days or VAN 125 mg qid for 10 days + BEZ 10 mg/kg IV once. The risk factors are: age > 65–70 years (most important), healthcare-associated CDI, prior hospitalization 3 months, continued non-CDI antibiotic use, and proton pump inhibitor started during/after CDI diagnosis. ^b^ Guideline established that first and second CDI recurrence are treated equally and subsequent recurrence as state on table. ^c^ SOC dosing: FDX 200 mg bid for 10 days or VAN 125 mg qid for 10 days or MET 500 mg tid for 10 days. ^d^ Only if vancomycin and fidaxomicin are not available. ^e^ Severe CDI is defined heterogeneously between guidelines: white blood cell count 15,000 cells/mL or serum creatinine >1.5 mg/dL (for IDSA/SHEA); one of the following factors at presentation: temperature > 38.5 °C, white cell count > 15,000 cell, and rise in serum creatinine > 50% above the baseline (for ESCMID); clinical judgment accounting risk factors: previous CDI, age > 65 years, body temperature > 38.5 °C, 10 or more bowel movements within 24 h, severe abdominal pain due to CDI, white blood cell > 15,000 cells/mL, serum creatinine > 1.5 mg/dL or an increase of 50% greater than baseline, presence of active malignancy, or albumin < 2.5 mg/dL (for Taiwan Guidelines); and serum albumin < 3 g/dL plus either white blood cell count 15,000 cells/mL or abdominal pain (for Mexican Consensus). ^f^ IV tigecycline could be given as adjunctive therapy if patient is unable to take oral medication, deteriorating or progressing to fulminant CDI. ^g^ FDX EPX: 200 mg bid PO for 5 days, then 200 mg qod PO for 20 days (on day 7–25). ^h^ Risk factors: ongoing antibiotic use, prior episode of CDI, age > 65 years, severity of disease, use of proton-pump inhibitor, females, immunocompromising conditions such as inflammatory bowel disease, solid-organ transplantation, chemotherapy, chronic kidney disease, hypogammaglobulinemia, or CDI caused by ribotype 027. ^i^ Vancomycin tapered/pulsed regimen: 125 mg 4 times daily for 10–14 days, 2 times daily for 7 days, once daily for 7 days, and then every 2 to 3 days for 2 to 8 weeks. ^j^ Patient with risk factors can receive same CDI therapy as patient without risk factors if no prior received VAN, FDX, or TEC. ^k^ Vancomycin extended regimen: 125 mg qid PO for 14 days, then 125 mg bid PO for 7 days, 125 mg qd PO for 7 days, 125 mg qod PO for 7 days, 125 mg q3d PO for 7–21 days (a total of 6–8 weeks). ^l^ Fulminant CDI is defined by hypotension, shock, ileus, and/or toxic megacolon. **Abbreviations:** CDI, *Clostridioides difficile* infection; IDSA/SHEA; Infectious Disease Society of America and Society for Healthcare Epidemiology of America; ESCMID, European Society of Clinical Microbiology and Infectious Disease; FDX, fidaxomicin; VAN, vancomycin; MET, metronidazole; TEC, teicoplanin; BEZ, bezlotoxumab; EPX, extended regimen; RAX, rifaximin; FMT, fecal microbiota transplantation; NaCl, normal saline; NGT, nasogastric tube; SOC, standard of care; PO, oral; IV, intravenous; tid, three times per day; qid, four times per day; bid, twice per day; qd, once per day; qod, every other day; q3d, every 3 days.

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
