# Peer review of "Best Practices in the Management of Clostridioides difficile Infection in Developing Nations"

_tropicalmed, 2024, doi:10.3390/tropicalmed9080185_

Round 1

Reviewer 1 Report

Comments and Suggestions for Authors

These studies analyzed the epidemiology of Clostridioides difficile in 29 countries in Asia, Africa and Latin America. The countries included in these studies have a Human Development Index (HDI) below 0.8. The authors also presented CDI diagnosis tests used in these countries, such as Toxin A/B latex agglutination, 16srDNA + Toxin A/B PCR and Toxin A/B Elisa. The authors present a therapeutic option for the treatment of CDI i.e. fecal microbiota transplantation, FMT, which is used in many developing countries as well as in Europe and US.

The authors provide information about that two Live Biotherapeutic Products (LBPs) designed to restore gut microbial balance and function have been approved by the U.S. Food and Drug Administration.

This article could be interesting to a wide range of recipients. In my opinion, the authors should consider adding information on the criteria for the diagnosis of CDI in the United States and European countries.

I think that adding CDI rates in the United States and European countries would allow comparison of CDI in developing and developing countries.

A very interesting article, which deals with many aspects of the diagnosis and treatment of CDI.

Author Response

Comments 1: These studies analyzed the epidemiology of Clostridioides difficile in 29 countries in Asia, Africa and Latin America. The countries included in these studies have a Human Development Index (HDI) below 0.8. The authors also presented CDI diagnosis tests used in these countries, such as Toxin A/B latex agglutination, 16srDNA + Toxin A/B PCR and Toxin A/B Elisa. The authors present a therapeutic option for the treatment of CDI i.e. fecal microbiota transplantation, FMT, which is used in many developing countries as well as in Europe and US. 

The authors provide information about that two Live Biotherapeutic Products (LBPs) designed to restore gut microbial balance and function have been approved by the U.S. Food and Drug Administration. 

This article could be interesting to a wide range of recipients. In my opinion, the authors should consider adding information on the criteria for the diagnosis of CDI in the United States and European countries.

Response 1: CDI diagnosis in the United States and Europe are presented in section 3.1 titled “State of the art in CDI diagnostics”.

Comments 2: I think that adding CDI rates in the United States and European countries would allow comparison of CDI in developing and developing countries.

Response 2: The CDI incidence rate for the U.S. is presented in the introduction section, and we have also added the most updated incidence density of a European report.

Reviewer 2 Report

Comments and Suggestions for Authors

I have read with interest the manuscript submitted by Mendo-Lopez et al.

I have a few comments to be addressed in order to improve the quality of the manuscript:

"Clostridioides difficile infection (CDI) is a well-known cause of hospital-acquired infectious diarrhea" - maybe C. difficile is the cause of HAIs, not CDI. Furthermore, infection should not be italicized.

Keep in mind that the abstract is the most read part of the article. This should be representative of your findings. In its current form, it seems it has been written in a rush, just to "check a box". 

All countries that fit in the inclusion criteria should be mentioned. No data from Europe?

row 53 - box 2 refers to abbreviations, not to what is indicated

in rows 127-144 - I highly suggest adding more recent studies on this issue, given it's dynamic nature and progressive interest accorded over the last years.

rows 200-206, 328 - I suggest using the latest, updated guideline on this issue; Moreover, an interesting and recent article on this issue would also be: https://www.ncbi.nlm.nih.gov/pmc/articles/PMC9780550/ 

All in all, I consider that the manuscript has somehow failed my expectations. There is limited and not systematized information about the developing countries in the beginning (with a biased selection of countries/territories to analyze), and later a lot of already known, general information.

I recommend deleting the information that does not fit into the chosen topic (maybe perform a parallel review that could contain them) and focusing on this one, in a more organized and comprehensive manner. Given the nature of the manuscript - review - you must include all / most of the information on this topic, not being selective.

The reference list is not edited according to the mdpi pattern.

Comments on the Quality of English Language

minor

Author Response

Comment 1: "Clostridioides difficile infection (CDI) is a well-known cause of hospital-acquired infectious diarrhea" - maybe C. difficile is the cause of HAIs, not CDI. Furthermore, infection should not be italicized.

Response 1: Observations were addressed in the manuscript. Thanks for the recommendation.

Comment 2: Keep in mind that the abstract is the most read part of the article. This should be representative of your findings. In its current form, it seems it has been written in a rush, just to "check a box". 

Response 2: We have edited the abstract and focused on the information presented in the review.

Comment 3: All countries that fit in the inclusion criteria should be mentioned. No data from Europe?

Response 3: Europe mainly includes developed countries. However, we have found a few countries that fulfilled our definition of developing countries and added all available data in a new section (2.1.4 Europe).

Comment 4: row 53 - box 2 refers to abbreviations, not to what is indicated

Response 4: Observation was addressed in the manuscript

Comment 5: in rows 127-144 - I highly suggest adding more recent studies on this issue, given it's dynamic nature and progressive interest accorded over the last years.

Response 5: We did not find any further specific data in developing countries about this topic. However, we added reference 31 that is a most update information regarding this topic in a developed country. If reviewer has any specific study, we are eager to add the information in this section.

Comment 6: rows 200-206, 328 - I suggest using the latest, updated guideline on this issue; Moreover, an interesting and recent article on this issue would also be: https://www.ncbi.nlm.nih.gov/pmc/articles/PMC9780550/ 

Response 6: We appreciated the suggestion, and the information was updated in the section by adding references 76 (updated 2021 IDSA guideline) and 77 (suggested article).

Comment 7: All in all, I consider that the manuscript has somehow failed my expectations. There is limited and not systematized information about the developing countries in the beginning (with a biased selection of countries/territories to analyze), and later a lot of already known, general information.

I recommend deleting the information that does not fit into the chosen topic (maybe perform a parallel review that could contain them) and focusing on this one, in a more organized and comprehensive manner. Given the nature of the manuscript - review - you must include all / most of the information on this topic, not being selective.

Response 7: From the beginning, this manuscript was a narrative review, and we do not intend to perform a systematic review. Regarding bias selection, we used an HDI cut-off for our definition of developing countries. HDI classified all countries in four different categories (very-high, high, medium, and low). The first one corresponds to an HDI >= 0.8 and we considered all included countries as developed and all the other 3 categories as developing countries.

Comment 8: The reference list is not edited according to the mdpi pattern.

Response 8: We appreciated the observation about the references, and we have adjusted them to MDPI pattern.

Reviewer 3 Report

Comments and Suggestions for Authors

This article is a review of best practices in managing Clostridioides difficile infection (CDI) in developing countries, covering epidemiology, diagnosis, management, and prevention of CDI in these regions. The article is well-structured and logically organized, but there are several issues that need to be addressed:

1.       The introduction provides insufficient information about the molecular layer of Clostridioides difficile.

2.       The seriousness of Clostridioides difficile infection is not adequately emphasized in the introduction.

3.       The countries listed in Table 1 do not include all developing countries; it should be revised to include only some developing countries.

4.       The abrupt transition of epidemiology and pathology to the second chapter is awkward.

5.       Consider subdividing the sections further, such as CDI testing, outbreak detection, and prevention measures specific to developing countries, to provide better assistance.

6.       The article lacks emphasis on best practices; it vaguely lists treatment methods and their drawbacks without offering clear guidance.

7.       Consider including figures to illustrate the prevalence of CDI in different developing countries, diagnostic accuracy, and other key indicators to enhance the article's clarity and comprehensibility.

8.       Tables 1 and 2 in the article are not sufficiently clear and concise.

Author Response

Comments:

  1. The introduction provides insufficient information about the molecular layer of Clostridioides difficile.
  2. The seriousness of Clostridioides difficile infection is not adequately emphasized in the introduction.

Response: Information regarding molecular layer and seriousness of the disease were added in the introduction section. We appreciate the recommendations.

  1. Consider subdividing the sections further, such as CDI testing, outbreak detection, and prevention measures specific to developing countries, to provide better assistance.

Response:

We addressed CDI testing on section 3.1 by comparison of main CDI guidelines with developing countries guidelines/ studies. Additionally, we do not intend to give specific recommendations in CDI diagnosis, treatment and preventive measures. However, we presented on section 4.5 (“Opportunity: Enhanced Antimicrobial Stewardship”) where we presented different studies including developed and developing countries and explained the positive impact in CDI with its implementation.

Finally, we did not find significant studies reporting outbreaks for developing countries. For this reason, we did not consider in developing a full section for this topic.

Comment: 

  1. The article lacks emphasis on best practices; it vaguely lists treatment methods and their drawbacks without offering clear guidance.

Response: There are no clear guidelines for CDI management in developing countries. In this narrative review, we gather the published data for developing countries and presented for the readers. We do not intend to provide any specific guidance for diagnosis and management in developing countries. We aim to provide in all information available, so physician can guide their CDI diagnosis and treatment based on their medical judgment and healthcare resources.

Comments:

  1. The countries listed in Table 1 do not include all developing countries; it should be revised to include only some developing countries.
  2. The abrupt transition of epidemiology and pathology to the second chapter is awkward.
  3. Consider including figures to illustrate the prevalence of CDI in different developing countries, diagnostic accuracy, and other key indicators to enhance the article's clarity and comprehensibility.
  4. Tables 1 and 2 in the article are not sufficiently clear and concise.

Response: We appreciate all these recommendations. Regarding Table 1, we consider moving it to Supplemental Material to make it more readable throughout the review but keep it available for the readers to review any specifics of each included studies. Instead, we present Figure 1 that show the epidemiology distribution of the developing countries. Regarding Table 2, we made it more concise to improve reading.

Round 2

Reviewer 2 Report

Comments and Suggestions for Authors

I appreciate the author's efforts in addressing my comments. The quality of the manuscript has significantly improved.

My only minor remarks would be the resolution of Figure 1, which is very low and Table 2 does not have all abbreviations defined (E.g. BEZ).

Some more schematic representation (within a figure) of data from Table 2 would have been great, if possible.

Best regards,

Comments on the Quality of English Language

minor

Author Response

Comment 1: My only minor remarks would be the resolution of Figure 1, which is very low and Table 2 does not have all abbreviations defined (E.g. BEZ).

Response 1: We have increased the resolution of Figure 1 and added all abbreviations missing on Table 2.

Comment 2: Some more schematic representation (within a figure) of data from Table 2 would have been great, if possible.

Response 2: Table 2 compares different available guidelines in developing countries, the USA, and Europe, showing some differences between each other. For this reason, elaborating a schematic representation would not be possible as we provide available data and do not intend to provide specific recommendations for CDI management.

Reviewer 3 Report

Comments and Suggestions for Authors

it may be suitable for publication

Author Response

Comment 1: it may be suitable for publication

Response 1: We truly appreciate all the comments given. Thanks